

# High resolution EPICA ice core dust fluxes: intermittency, extremes and Holocene stability

Shaun Lovejoy[1], Fabrice Lambert[2]

[1]Physics Department, McGill University, 3600 University st., Montreal, Que. H3A 2T8, Canada
[2]Department of Physical Geography, Pontificia Universidad Catolica de Chile, Vicuna Mackenna 4860, Santiago, Chile

**Abstract.**

Recent research in climate variability as a function of temporal or spatial scale has shown that the majority of the variance power lies in what has up until now been considered an unimportant background with relatively little power in well-known frequencies, such as daily, seasonal, or orbital oscillations. Atmospheric variability as a function of scale can be divided in various dynamical "regimes" with alternating increasing and decreasing fluctuations: weather, macroweather, climate, macroclimate, megaclimate. Although a vast amount of data is available at small scales, the larger picture is not well constrained due to the scarcity and low resolution of long paleoclimatic time-series. Here, we analyse a unique centimetric resolution dust flux series from the EPICA Dome C ice-core in Antarctica that spans the past 800,000 years. The temporal resolution is 5 years over the last 400 kyrs, and 25 years over the last 800kyrs, enabling the detailed statistical analysis and comparison of eight glaciation cycles. The main spectral peak of the complete record is superposed on a scaling (power law) process and accounts for only 4-15% of the variability, the rest being in the scaling continuum, thus inverting the classical notions of foreground and background processes.

We analyzed the glacial-interglacial cycles using two definitions: a fixed duration of 100 kyrs (segments) and a variable duration defined by the interglacial dust minima (cycles). Segments and cycles were further divided into eight consecutive "phases". We found that the first two phases of each segment or cycle showed particularly large macroweather to climate transition scale $\tau_c$ (around 2 kyrs), whereas later phases feature centennial transition scales (average of 250 kyr). This suggests that interglacials and glacial maxima are exceptionally stable when compared with the rest of a glacial cycle. The Holocene (with $\tau_c \approx 4$ kyrs) had a particularly large $\tau_c$ but it was not an outlier when compared with the phase 1 and 2 of other cycles. For each phase we quantified the drift, intermittency, and extremeness of the variability. Phases close to the interglacials (1, 2, 8) show low drift, moderate intermittency, and strong extremes, while the "glacial" middle phases 3-7 display strong drift, weak intermittency, and weaker extremes.

## 1 Introduction

Over the early Pleistocene - until about 800 kyrs ago – marine sediment paleotemperature reconstructions exhibit strongly periodic behaviour (Lisiecky & Raymo 2005 Paleoceanography). At (obliquity) frequencies of (41 kyrs)[-1], spectra





show a strong peak that rises sharply above the spectral "background" by a factor of more than 10 (Huybers, 2007). In contrast, in the more recent period, while there is still a fairly cyclic advance and retreat of ice sheets - the "ice ages" – the phenomenon is by no means perfectly periodic with the glacial maxima spaced at roughly 100 kyrs intervals (Jouzel et al., 2007). The corresponding spectral peak appears to be so broad that simple interpretation in terms of Milankovitch forcings

and responses are not obvious. It has even been suggested (Lovejoy, 2015) that an alternative theorization is of a transition between two scaling regimes: from the climate to macroclimate. While such a broad transition may still ultimately be astronomical in origin, it can no longer be easily viewed as a simple spectral spike overlying an otherwise uninteresting and unimportant background.

One of the difficulties in characterizing the variability is that the most reliable paleo-temperature reconstructions

(from ice core isotope ratios), rapidly lose their resolutions as we move to the bottom of the ice column. Fig. 1 shows this visually for the EPICA Antarctic ice core (5787 measurements in all), the curve becomes noticeably smoother as we move back in time. In terms of data points, the more recent 100 kyr period has more than 3000 points (≈30 year resolution) whereas the most ancient 100 kyr period has only 137 (≈800 year resolution). This implies that while the most recent glacial-interglacial cycle can be perceived with reasonable detail, it is hard to quantitatively compare it to previous cycles - or to

deduce any general cycle characteristics.

Following the influential paper by (Mitchell, 1976), the classical approach assumes that there is an unimportant (more or less) white noise "background" upon which are superimposed significant quasi-oscillatory processes. However, using modern and paleoclimatic data analyzed by spectral methods or by using a more transparent approach, fluctuation analysis discussed below, (Lovejoy, 2015) argued that on the contrary, almost all the variability is in the "background" pointing out

that Mitchell's early picture greatly differed from the variability determined by modern paleo-data.

Fluctuation analysis (Lovejoy and Schertzer, 2013; Nilsen et al., 2016), gives a relatively simple picture of atmospheric variability (Fig. 2). The figure shows a series of regimes each with variability alternately increasing and decreasing with scale. From left to right we see weather scale variability, in which fluctuations tend to persist, building up with scale - they are unstable - increasing up to the lifetime of planetary structures (about 10 days), followed by a

macroweather regime with fluctuations tending to cancel each other out, decreasing with scale, displaying stable behaviour. In the last century, anthropogenically forced temperature changes dominate the natural (internal, macroweather) variability at about 10- 20 years; in pre-industrial  periods the lower frequency climate regime starts at scale $\tau_c$ between somewhere 100 and 1000 years.  Further to the right of Fig. 2 we can see the broad peak associated with the glacial cycles at about 50kyrs (half the 100 kyr period) and then at very low frequencies, the megaclimate regime again shows increasing variability with

scale.  In between the climate and megaclimate regimes, the fluctuations decrease with scale over a fairly short range: from about 100 kyrs to 500 kyrs.

Figure 2 shows the average fluctuation behaviour, but this can potentially hide large variations from epoch to epoch. Of particular importance is the macroweather - climate transition scale $\tau_c$. A particularly large $\tau_c$ implies that temperatures were stable over long times and some (e.g. (Petit et al., 1999)) have argued that the exceptional Holocene stability (inferred



from the Antarctic Vostok core) allowed the development of farming and civilisation. Others (Berner et al., 2008) argued the case of Holocene instability on the basis of paleo SST records. In this paper, we use a unique high resolution and long paleo record of dust fluxes to attempt to answer this question.

In this work, we focus on the EPICA Dome C dust flux record, which has a 55 times higher resolution than the temperature record, including high resolution over even the oldest cycle (Lambert et al., 2012). Antarctic dust fluxes are generally correlated with temperature, but are also affected by climatic conditions at the source and during transport (Lambert et al., 2008; Maher et al., 2010). The analysis of the dust record presented here can therefore be thought of as a more holistic climatic parameter that includes not only temperature changes, but describes atmospheric variability as a whole (including wind strength and patterns, and the hydrological cycle).

## 2 The data

Unlike oxygen isotopes that diffuse and lose their temporal resolution at high pressures and densities, the relatively large dust particles diffuse much less and have recently been used to estimate the dust flux over every centimetre of the 3.2 km long EPICA core (298,203 measurements, (Lambert et al., 2012)). The temporal resolution of this series varies from 0.81 years to 11.1 yrs (the averages over the most recent and the most ancient 100 kyrs respectively). The worst temporal resolution of 25 years per centimeter occurs around 3050 m depth, with the result that at that resolution, there are virtually no missing data points (Fig. 1).

Fig. 3a shows a succession of 10 factors of 2 "blowdowns" (upper left to lower right), or equivalently, 10 blow-ups (lower right to upper left; 11 different resolutions). In order to avoid smoothing, the data was "zoomed" in depth rather than time, but the point is clear: the signal is very roughly scale invariant, at no stage is there any sign of obvious smoothing, and the quasi-periodic 100 kyr oscillations is the only obvious time scale (we quantify this below). In comparison with more common paleoclimate signals such as temperature proxies, the dust flux itself is already quite spiky. However, it also displays spiky transitions: in Fig. 3b we can note the strong spikiness associated with strongly non Gaussian variability: the intermittency. This figure shows the mean absolute difference between neighbouring values normalized by the average over each 290 point long segment. At each resolution, the solid red line indicates the maximum spike expected if the process was Gaussian, and the upper dashed lines the expected level for a (Gaussian) spike with probability $10^{-6}$. Again, without sophisticated analysis, we can see that the spikes are wildly non Gaussian, frequently exceeding the $10^{-6}$ level even though each segment has only 290 points, with the spikiness being nearly independent of resolution (for more on spike plots, see (Lovejoy, 2018).

How can we comprehend variability over such wide ranges and with such sharp spikiness? Due to its link to temperature, wind, and precipitation, Antarctic dust data is representative of the climatic state of the atmosphere at the hemispheric level. At large scales, the ice-core dust record is mostly representative of temperature variability, while the influence of wind and precipitation variability should become progressively stronger with smaller scales.





Figure 4 shows various spectral analyses. The figure superposes the spectra of the 25 year resolution flux over the full 800 kyr period (red) and the spectrum of the log of the flux (blue), each averaged over ten logarithmically spaced bins up to frequencies of (1kyr)$^{-1}$, along with the average spectrum of the 5 year resolution data (green over the last 400 kyrs). For the latter, the periodograms of each the four most recent 100 kyr cycles were averaged, but the full spectral resolution (5yrs)$^{-1}$

was retained. Since this is a log-log plot, power laws appear as straight lines. Also shown in the figure are fits to the bi-scaling function

$$E(\omega) = \frac{a}{\left(\omega/\omega_c\right)^{\beta_h} + \left(\omega/\omega_c\right)^{\beta_l}}$$

that smoothly transitions between a spectrum with $E(\omega) \gg \omega^{-\beta_h}$ at $\omega > \omega_c$ and $E(\omega) \gg \omega^{-\beta_l}$ at $\omega < \omega_c$. The figure shows the regressions with $\beta_l = -2.5$, $\beta_h = 1.7$, and a = 7.5 (mg/m$^2$/yr)$^2$yr, $\omega_c \approx$ (145kyrs)$^{-1}$ for the fluxes, and a = 0.375

(mg/m$^2$/yr)$^2$yr, $\omega_c \approx$ (160kyrs)$^{-1}$ for the logarithms of fluxes. According to the figure, the high frequency climate regime scaling continues to about (250 yrs)$^{-1}$ before flattening to a very high frequency scaling ($\beta \approx 0.4$) "macroweather" regime (Lovejoy and Schertzer, 2013).

The plot graphically counterposes two views of the variability. Although we clearly see a spectral maximum at around (100 kyrs)$^{-1}$, the broad bispectral scaling model already accounts for 96% of the spectral energy (variance) leaving

only 4% for the (extra) contribution from the (near) (100kyrs)$^{-1}$ Milankovitch frequency. If it is argued that the logarithm of the flux is more physically relevant - justifying taking the logarithms (blue) - the situation is barely changed. Alternatively, we may take a narrow spectral spike model that approximates the spectral spike near (100 kyr)$^{-1}$ as a Gaussian shaped profile. With this model, the spike is localised at (94±0.9 kyrs)$^{-1}$ and contributes a total of 31% of the total variance, however, not all of this is above what we would expect from a scaling background; the exact amount depends on how the

background is defined. For example, over the range from the 6$^{th}$ to the 11 highest frequencies in this discrete spectrum (from (133 kyrs)$^{-1}$ to (72 kyrs)$^{-1}$), in comparison to the background over this range, there is an enhancement of about 80% due to the strong peaks (the enhancement is about 100% for the 7$^{th}$ to the 12$^{th}$ frequencies). This means although the (94±0.9 kyr)$^{-1}$ peak represents 31% of the total variability over the range from (800 kyrs)$^{-1}$ to (25 yrs)$^{-1}$, it is only about 15% above the "background" (note that only 5% of the total variance is between (25yrs)$^{-1}$ and (1 kyr)$^{-1}$). The overall conclusion is that the

background represents between 85% and 96% of the total variance.

Scaling is a statistical symmetry - here, it says that *on average* the statistics at small, medium and large scales are the same in some way. The difficulty is that on a single realization – such as that available here, a single core from a single planet earth – the symmetry will necessarily be broken. For example, in the spectrum Fig. 4, in each of the proposed scaling regimes, scaling only predicts that the actual spectrum from this single core will vary about the indicated straight lines that

represent the ensemble behaviour. Since this variability is strong, we made the potential scaling regimes more obvious by



either averaging the spectrum over frequency bins (the red and blue spectra) – or by breaking the series into shorter parts and averaging the spectra over all the parts, effectively treating each segment as a separate realization of a single process (green).

This already illustrates the general problem: in order to obtain robust statistics we need to average over numerous realizations – and since here we have a single series, the best we can do is to break the series into disjoint segments and average the statistics over them. Yet at the same time, in order to see the wide-range scaling picture (which also helps to more accurately estimate the scaling properties/exponents), we need segments that are as long as possible. The compromise that we chose between numerous short segments and a small number of long ones was to break the series into 8 glacial-interglacial cycles, and each cycle into 8 successive phases. As a first approximation, we defined eight successive 100kyr periods (hereafter called "segments", Fig. 5, top set), corresponding fairly closely to the main periodicity of the series. As we discussed, the spectral peak is broad implying that the duration of each cycle is variable – the cycles are only "quasi-periodic". It is therefore of interest to consider an additional somewhat flexible definition of cycles defining them as the period from one interglacial to the next (hereafter called "cycle", Fig. 5, bottom set). The break points were taken at interglacial optima: 0.4, 128.5, 243.5, 336, 407.5, 490, 614, 700, 789 kyrs BP, i.e. 96.9±18.7 kyrs per cycle. Using this latter definition, the cycles were nondimensionalized so that nondimensional time was defined as the fraction of the cycle, effectively stretching or compressing the cycles by ±19%.

With either of these definitions, we have 8 segments or cycles, each with 8 phases. Fig. 6 shows the phase by phase information summarized by the average flux over each cycle including the dispersion of each cycle about the mean (for the segments in the top set, and the cycles in the bottom set). We see that the variability is highest in the middle of a cycle and lowest at the ends.

## 3 Results and discussion

### 3.1 Full Dataset Analysis

In order to proceed to a further quantitative analysis of the types of statistical variability, and of the macroweather-climate transition scale, we need to make some definitions. We have discussed a commonly used way of quantifying fluctuations: Fourier analysis. It quantifies the contribution of each frequency range to the total variance of the process (Fig. 4). However, the interpretation of the spectrum is neither intuitive, nor straightforward. We already mentioned that Mitchell's classical spectral paradigm implied a massive underestimate of the variability; it turns out that the highly non-Gaussian spikiness (e.g. Fig. 3b), implies strong Fourier space spikes; indeed, (Lovejoy, 2018) found that the probability distribution of spectral amplitudes can themselves be power laws. This has important implications for interpreting spectra, especially those estimated from single series ("periodograms"): if the spectral amplitudes are highly non-Gaussian, then we will typically see strong spectral spikes that are purely random in origin. This makes it very tempting to attribute quasi-oscillatory processes to what are in fact random spectral peaks. A final reason for considering the real (rather than Fourier) space variability (fluctuations) is that the spectrum is a second order statistical moment (the spectrum in the Fourier



transform of the autocorrelation function). While second order moments are sufficient for characterizing the variability of Gaussian processes, in the more general and usual case - especially with the highly variable dust fluxes - we need to quantify statistics of higher orders. In particular, the higher order statistics that characterize the extremes. Here, we will use two simple concepts to describe variability and intermittency (or spikiness) of the data. The basic tool is the Haar fluctuation

which is simply the absolute difference of the mean over the first and second halves of an interval:

$$\Delta F\left(\Delta t\right) = \frac{2}{\Delta t} \int_{t-\Delta t/2}^{t} F(t')dt' - \frac{2}{\Delta t} \int_{t-\Delta t}^{t-\Delta t/2} F(t')dt'$$

We can characterize the fluctuations by their statistics. For example, by analyzing the whole dataset using intervals of various lengths, we can thus define the variability as a function of scale (i.e. interval length). If over a range of time scales $\Delta t$, there is no characteristic time, then this relationship is a power law, and the mean absolute fluctuation varies as:

$$\left\langle \left| \Delta F\left(\Delta t\right) \right| \right\rangle \propto \Delta t^{H}$$

where "< >" indicates ensemble average, here an average over all the available disjoint intervals. A positive H implies that the average fluctuations increase with scale. This situation corresponds to unstable behavior identified with the climate regime. In contrast, when H is negative, variability converges towards a mean state with increasing scale. This is the situation found in the stable macroweather regime.

More generally, we can consider other statistical moments of the fluctuations, the "generalized structure functions", $S_q(\Delta t)$:

$$S_{q}\left(\Delta t\right) = \left\langle \left| \Delta F\left(\Delta t\right) \right|^{q} \right\rangle \propto \Delta t^{\xi(q)}$$

If the fluctuations are from a Gaussian process, then their exponent function is linear: $\xi(q) = qH$. More generally

however, $\xi(q)$ is concave and it is important to characterize this, since the nonlinearity in $\xi(q)$ is due to intermittency, i.e. sudden, spiky transitions (for more details on Haar fluctuations and intermittency we refer to (Lovejoy and Schertzer, 2012)). We therefore decompose $\xi(q)$ into a linear and a nonlinear (convex) part K(q), with K(1)=0:

$$\xi\left(q\right) = qH - K\left(q\right)$$

A simple way to quantify the intermittency is thus to compare, the mean and Root Mean Square (RMS) Haar fluctuations:

$$S_{1}\left(\Delta t\right) = \left\langle \left| \left(\Delta F\left(\Delta t\right)\right) \right| \right\rangle \propto \Delta t^{\xi(1)} = \Delta t^{H} \qquad (10)$$

$$S_{2}\left(\Delta t\right)^{1/2} = \left\langle \left(\Delta F\left(\Delta t\right)\right)^{2} \right\rangle^{1/2} \propto \Delta t^{\xi(2)/2} = \Delta t^{H-K(2)/2}$$



with ratio:

$$S_1(\Delta t)/S_2(\Delta t)^{1/2} = \left\langle \left|\Delta F(\Delta t)\right| \right\rangle / \left\langle \left(\Delta F(\Delta t)\right)^2 \right\rangle^{1/2} \propto \Delta t^{K(2)/2}$$

where we estimate $S(\Delta t)$ using all available disjoint intervals of size $\Delta t$. These expressions are valid in a scaling regime. Since the number of disjoint intervals decreases as $\Delta t$ increases, so does the sample size, hence the statistics are less reliable

at large $\Delta t$, explaining the somewhat "noisy" appearance of plots such as Fig. 7a that shows the result when $S_1(\Delta t)$ and $S_2(\Delta t)^{1/2}$ are calculated over the entire series at 25 year resolution. The only way to completely quantify this effect is with a stochastic model of the process.

Figure 7a shows that Haar fluctuations have simple interpretations in terms of the variability of the dust flux. For example, typical variations over a glacial-interglacial cycle (half cycle ≈ 50 kyrs) are about ±3mg/m²/yr (dashed line). From

the figure we see there is a short regime with $H<0$ (up to about 250 yrs), a scaling regime with $H>0$ (up to glacial-interglacial periods (≈50 kyrs) and finally a long time scale decrease in variability that is possibly (but not obviously) scaling. As expected, the regimes correspond to those indicated in Fig. 4 with the relation $\beta=1+\xi(2)$ where $E(\omega) \approx \omega^{-\beta}$ and represent macroweather, climate, and macroclimate, respectively.

If the intermittency is small ($K(q) \approx 0$), then $\xi(2)/2 \approx H$ and $S_2(\Delta t)^{1/2} \propto S_1(\Delta t) \propto \Delta t^H$ so that the lines in Fig. 7a

will be parallel. From the figure, we notice the lines slowly converge; the bottom line in Fig. 7a shows the ratio $S_1(\Delta t)^2/S_2(\Delta t) \propto \Delta t^{2\xi(1)-\xi(2)} \propto \Delta t^{K(2)}$. This non-Gaussian behaviour is a symptom of intermittency. As a first – and simple – quantification of the rate of convergence (and hence intermittency) we can take the variance ratio $S_1(\Delta t)^2/S_2(\Delta t)$ which is also shown in the figure (whose exponent is $2\xi(1)-\xi(2) = K(2)$). A useful approximate relationship is $K(2) \approx 2C_1$, so that the variance ratio indicates that over the intermediate (climate) regime $C_1 \approx 0.05$, with a

hint of a transition to a higher intermittency regime with $C_1 \approx 0.15$ over the range of scales 10-100 kyrs.

For theoretical reasons, it turns out that the intermittency near the mean (q=1) is best quantified by the parameter $C_1 = K'(1)$. Since $K(1) = 0$, it turns out that the ratio exponent $K(2)/2 \approx C_1$, so that the slope of the bottom curve in Fig. 7a is ≈$2C_1$.

As expected from the spectra, there are three regimes in the dust data (Fig. 7a). The first (high frequency) regime on the left shows fluctuations slowly decreasing to about $\tau_c \approx 250$ years (with $\xi(1) = H \approx -0.05$) beyond which the fluctuations

start to increase with scale with $\xi(1) = H \approx 0.37$. Finally, at scales beyond a broad peak at around 50 kyrs, the amplitude of the fluctuations rapidly falls corresponding to the far left (low frequency) regime of the spectrum (Fig. 4). The same Haar analysis of the ice ages with nondimensional time, averaged over all the ice ages, yields similar results.





The overall behavior for variability and intermittency, at high resolution (5 yrs) over the last 4 glacial-interglacial cycles can be seen in Fig. 7b and 7c, respectively. While the mean to RMS ratio in Fig. 7a is intuitive, a more accurate estimate of $C_1$ uses the intermittency function $G(\Delta t)$:

$$G\left(\Delta t\right)=\left\langle \Delta F\right\rangle \left[\frac{\left\langle \Delta F^{1-\Delta q}\right\rangle}{\left\langle \Delta F^{1+\Delta q}\right\rangle}\right]^{1/(2\Delta q)} \propto \Delta t^{\xi(1)-\xi'(1)} = \Delta t^{C_1}; \quad \Delta q \to 0$$

whose exponent is $C_1$; we use this in fig. 7c.

From Fig. 7a-c, we see that all cycles display a very similar behavior. Also, due to the higher resolution in Fig. 7b, the macro-weather to climate transition time $\tau_c$ at centennial time scales is clearer. Fig. 7c shows the corresponding figure for $G\left(\Delta t\right) \approx \Delta t^{C_1}$, showing that the intermittency also has consistent cycle to cycle behaviour, although it displays a transition

at scales of thousands of years; several times longer than for the fluctuations (there is no necessity for the intermittency transition to be at the same scale as for $S_1$).

The intermittency exponent $C_1$ quantifies the *rate* at which the clustering near the mean builds up as a function of the range of scales over which the dynamical processes act, it only partially quantifies the spikiness. For this we need other exponents, in particular the exponent $q_D$ that characterizes the tails of the probability distributions. This is

because scaling in space and/or time generically gives rise to power law probability distributions (Mandelbrot, 1974; Schertzer and Lovejoy, 1987). Specifically, the probability ($Pr$) of a random dust flux fluctuation $\Delta F$ exceeding a fixed threshold $s$ is:

$$\Pr\left(\Delta F > s\right) \approx s^{q_D}; \quad s \gg 1$$

Where the exponent $q_D$ characterizes the extremes (for example, $q_D \approx 5$ has been estimated for wind or temperature

(Lovejoy and Schertzer, 1986) and $q_D = 3$ for precipitation (Lovejoy et al., 2012). A qualitative classification of probability distributions describes classical exponential tailed distributions (such as the Gaussian) as "thin tailed", log normal (and log-Levy) distributions as "long-tailed", and power law distributions as "fat tailed". Whereas thin and long tailed distributions have convergence of all statistical moments, power distributions only have finite moments for orders $q < q_D$.

Fig. 8 shows the fluctuation probabilities of the entire 800 kyr series at 25 year resolution. We see that the large

fluctuations (the tail) part of the distribution is indeed quite linear on a log-log plot with exponents $q_D \approx 2.75$ and $2.98$ in time and depth respectively (both fit to the extreme 0.1% of the distributions). To get an idea of how extreme these distributions are, consider the depth distribution with $q_D = 2.98$. With this exponent, dust flux fluctuations 10 times larger than typical fluctuations occur only $10^{2.98} \approx 1000$ times less frequently. In comparison, for a Gaussian, they would be $\approx 10^{23}$

times less likely; they would never be observed. Similarly, these extremes are much stronger than log-normals that are sometimes invoked in this context.

These power law fluctuations are so large that according to the classical assumptions, they would be outliers. While Gaussians are mathematically convenient and can be justified when dealing with measurement errors, in atmospheric science

thanks to the scaling, very few processes are Gaussian. Extremes occur much too frequently, a fact that has been regularly underscored starting in the 1980's (for a review see table 5.1a,b. in (Lovejoy and Schertzer, 2013)).

At best, Gaussians can be justified for additive processes, with the added restriction that the variance is finite. However, once this restriction is dropped, we obtain "Levy distributions" with power law extremes, but with exponents $q_D$<2. The Gaussian assumption also fails for the additive but scaling $H$ model (Lovejoy, 2015; Lovejoy and Mandelbrot,

1985). Most importantly, Gaussians are irrelevant for multiplicative processes: these generally lead to power law extremes but without any restriction on the value of $q_D$ (Mandelbrot, 1974; Schertzer and Lovejoy, 1987). Note that multiplicative random *variables* lead to somewhat less extreme log Levy and lognormal distributions (i.e. the logarithms are Levy or Gaussian (Aitchison and Brown, 1957)).

To underscore the importance of nonclassical extremes, (Taleb, 2010) introduced the terms "grey and black swans".

Originally, the former designated Levy extremes, and the latter was reserved for extremes that were so strong that they were outliers with respect to any existing theory. However, the term "grey swan" never stuck, and the better-known expression "black swan" is increasingly used for any power law extremes.

All of this is important in analyzing the dust flux series where extreme events may be associated with qualitatively new phenomena such as Dansgaard-Oescger events or even tipping points. The existence of black swan extremes leads to a

conundrum: since black swans already lead to exceptionally big extremes, how can we distinguish "mere" black swans from potentially real outliers? In any event, for each empirical distribution, Fig. 8 shows the Gaussian with the same mean and standard deviation. We see that the empirical distributions are generally quite far from Gaussian.

### 3.2 Analysis by Phases

The spectrum (Fig. 4) quantitatively establishes the existence of a broad spectral peak at around $(100 \text{ kyrs})^{-1}$. In this

section, we will therefore profit from the high resolution nature of the data to examine the statistical properties of consecutive sections (hereafter called phases) of each glacial-interglacial cycles. Since the spectra showed that there were wide scale ranges that are scale invariant – power laws - we are interested in characterizing the scaling properties over the different phases. We have seen that there is relatively little cycle to cycle difference in statistical properties, (Fig. 7b,c). However, in Fig. 6 we saw that there appeared to be important differences depending on the phase of the cycle. Fig. 9

compares the statistics averaged over cycles and the statistics averaged over phases. The figure confirms that the phase to phase differences are much more important than the cycle to cycle differences. Particularly noticeable are the phase to phase

differences in the average fluctuations $\left\langle \left| \left( \Delta F(\Delta t) \right) \right| \right\rangle$ (lower left).





From the global statistics (e.g. Figs. 4, 7), it is clear that in each glacial-interglacial cycle there are two regimes, so that before characterizing the structure functions by their exponents (e.g. $H = \xi(1)$ for the mean fluctuations), we have to determine the macroweather-climate transition time scale $\tau_c$ whose average (from Fig. 4, 7) is about 250 years.

One way of estimating the transition scale $\tau_c$ is to make a bilinear fit of $\log_{10}S_1(\Delta t)$ (i.e. Haar with $q = 1$, the mean
absolute fluctuation) with the mean slopes -0.05 (small $\Delta t$) and slope +0.25 (large $\Delta t$; the values were chosen because they are roughly the $H$ estimates from the average over all the cycles) (Fig. 10). Bilinear fits were made for each phase of each segment (blue) as well as for each phase of each cycle (black). For each phase there were thus 8 transition scales, which were used to calculate the mean and its standard deviation, (shown here as representative black arrows). From the figure we see that at first (phases 1, 2) the transition scale is 1 – 2 kyrs, but that it rapidly moves to shorter ($\approx$250-400 yr) scales for the
other phases. The average transition scale over all phases is around 300 years. This is very similar to the behaviour shown in Fig. 7a lower right, calculated from the ensemble slopes (over all segments or cycles).

The figure shows that our results are robust since the results are not so different using dimensional and nondimensional time (segments and cycles). Comparing the blue and black curves we see that in all cases the early phases have much larger $\tau_c$ than the middle and later phases. Also shown in Fig. 10 (dashed) is a plot of the break points estimated
by a more subjective method that attempts to visually determine a break point on $\log S_1 - \log \Delta t$ plots. Again, we reach the same conclusion with quantitatively very similar results: a transition of millennia at the beginning, and a few centuries in the middle. The cycle average value ($\tau_c \approx 300$ years) is therefore not representative of the early phases where $\tau_c$ is many times larger (this includes the Holocene). The Holocene has an even larger transition scale ($\tau_c \approx 7.9$kyrs, X in Fig. 10), but it lies just outside the standard deviation of the first nondimensional phases (red arrows in Fig. 10), and is therefore not an outlier.
Rather than fix a phase and determine the variation of the mean fluctuation and intermittency function (Fig. 10), we can consider the variation of the Haar fluctuations at a fixed time scales and see how they vary from phase to phase (Fig. 11; for nondimensional time, cycle). The figure shows the phase to phase variation of Haar fluctuations at 50, 100, 200, 400, 800, 3300, 7000 years scales (bottom to top). Low H implies that the curves are bunched up, high H that they are well separated as the time interval $\Delta t$ is increased. For every time scale, there is a clear cyclicity (left to right), with fluctuation
amplitudes largest in the middle phases. We could note that the cycle to cycle variability is fairly large; about a factor of 2 (for clarity the error bars indicating the cycle to cycle spread were not shown).

Another useful characterisation of the phases is to directly consider the flux variability at a fixed reference scale, taken here as the 25 year resolution; quantifying the amplitude of the variability of each segment by its standard deviation $A$ at 25yr time scale (Fig. 12, lower left). This is not the *difference* between neighbouring values or fluctuation, it is rather the
variability of the series itself at 25 year resolution. For each of the phases, we have 8 estimates (one from each cycle); these are used to calculate the mean (black) and standard deviation shown by the error bars. We can see that the amplitude of the 25 yr scale fluctuations is about four times higher in the middle of the ice age (phase 4) than at the beginning (phase 1). The figure clearly shows the strong change of variability across the cycle.





We can also estimate the exponents $H$ and $C_1$. Figure 12 shows the result on the nondimensional phases of the range 600 years$<\Delta t<$5000 years, (upper left and right; the range was chosen to be mostly with $\Delta t>\tau_c$, and it was fixed so as to avoid any uncertainty associated with the algorithm used to estimate $\tau_c$). Recall that the fluctuation exponent $H>0$ quantifies the rate at which the average fluctuations increase with time scale. Similarly, the exponent $C_1$ characterizes the rate at which the

spikiness near the mean (the intermittency exponent) increases with scale. We see (upper left) that $H$ is small in the early phases (with H actually a little bit negative on average in the first phase due to the large $\tau_c$ value) with $H$ reaching a fairly high value in the later phases. $C_1$ on the other hand (upper right) decreases a bit in the middle the phases. The error bars show that there is quite a lot of cycle to cycle variability.

If $H$ quantifies the "drift" and $C_1$ the "spikiness", then Fig. 12 shows that the early phases have low drift and medium

spikiness, the middle phases have high drift and lower spikiness, while phases 5-8 have high drift but medium spikiness. To understand this better, consider the transition time scales in Fig. 10. The early 2 phases with the low drift and spikiness are also the phase with the longest transition scales. This means that the rate at which the variability builds up is small and that it only builds up over a short range of scales (from $\tau_c$ to roughly $\Delta t = 50$ kyrs, the half cycle duration, this can be checked on Fig. 9 that shows the phase by phase structure functions and intermittency functions). Conversely, the phases 3 and 4 with

high drift and high intermittency also have a smaller $\tau_c$ so that both the fluctuations and spikiness build up faster (Fig. 11) and over a wider range of scales (Fig. 10).

Whereas $C_1$ characterizes the intermittency near the mean, we have seen that probability exponent $q_D$ characterizes the extreme spikiness. Fig. 12 lower right, compares $q_D$ phase by phase. Recalling that small $q_D$ implies more extreme extremes, we see that the extremes are stronger in the beginning and end of the cycle, and somewhat less pronounced in the

middle phases of the cycle (note the overall mean is 2.62±0.42). Notice that for phase 8, $q_D$=2.03 (the mean); this is close to the value $q_D = 2$ below which the extremes are so strong that the variance (and hence spectrum) does not converge. An extreme (low) exponent $q_D$ phase implies that most of the time the changes in flux are small, but occasionally, there are huge transitions. Conversely, a high (less extreme) $q_D$ implies that there is a wider range of different flux changes so that most of the changes tend to be in a restricted range. We can now categorize the phase by phase spikiness as: extremes strong, and

medium spikiness (phases 1, 2, 8), and extremes intermediate and low spikiness (phases 3-7). For the cycle to cycle estimates (not shown), the value $q_D$ =2.75±0.41, seems to be fairly representative of all the cycles, although there is a slight tendency for $q_D$ to decrease for the older cycles implying that they may have been a bit more extreme than the recent ones.

## 4 Conclusions

Until now, a systematic comparison of the different glacial-interglacial cycles has been hindered by a limitation of the

most common paleoclimate indicators – the low resolution of temperature reconstructions from ice or ocean cores. Due to this intrinsic characteristic, the older cycles are poorly discerned; we gave the example of EPICA paleo temperatures whose resolution in the most recent cycle was 25 times higher than the resolution in the oldest one. In this paper, we therefore took



advantage of a unique dust flux dataset with 1 cm resolution measuring 320,000 cm. The most recent four cycles were discerned at 5 year resolution throughout (20,000 points per cycle) and the entire record of eight glaciations could be resolved at 25 year, and this, without signs of over-sampling or smoothing. At this resolution, each 100 kyr segment thus had 4000 data points allowing glacial-interglacial cycles to be quantitatively compared with each other.

Our spectral analysis established that the majority of the variability – 85 - 96% depending on the sharp or broad peak model - was in the macroweather and climate scaling regime "backgrounds" with an average transition scale of about 250 years. Therefore, the task of statistically characterizing the cycles reduced primarily to the problem of characterizing the transition scale $\tau_c$ and main exponents $H$, $C_1$, $q_D$ that characterize respectively the growth (or decay) of fluctuations, the intermittency with scale, and their extremes, and then comparing them over different cycles. Since experience has shown that
spectral analysis of climate series is not always easy to interpret, we characterized the variability using real space (Haar) fluctuations that can be directly interpreted in terms of differences or anomalies.

Since the glacial-interglacial cycles were only quasi periodic, we compared the simple fixed duration definition of 100 kyrs per cycle (which we call segments) with a slightly different nondimensional definition from interglacial to interglacial (which we call cycles), potentially better tailored to the dynamics. We then broke each segment and cycle into 8
consecutive phases; each 12.5 kyrs and (for the 25 year data) with 500 points with the Holocene corresponding to the most recent phase (for the nondimensional definition, the phases were similarly 1/8 of the cycle length). The main conclusions were that cycle to cycle variability of dust fluxes were relatively small compared to the systematic phase to phase variations across each cycle. In general, the conclusions were robust with respect to different cycle definitions (dimensional, non-dimensional).

Using various techniques, $\tau_c$ was found to be systematically larger in the early two phases than in the middle and later phases; about 2 kyrs but with nearly a factor of 4 cycle to cycle spread and equal to 300 years (with a factor of 2 spread) for the six remaining phases (Fig 10). Since the Holocene $\tau_c$ was found to be ≈ 3 - 5 kyrs, it would have been an outlier when compared to the overall cycle average, but - since it is in phase 1 - it was not an outlier with respect to the typically large phase 1 and 2 values. Similarly, the typical (RMS) variation in flux amplitude was smaller in the early phase increases by
(on average) a factor of 4 from ±0.13 mg/m²/yr to about ±0.5 mg/m²/yr in the middle and later phases. The Holocene (with an amplitude of ±0.08 mg/m²/yr) was again particularly stable with respect to the phase 1 of other cycles, but it was not an outlier.

Finally, if we characterize the phases' variability by $H$, $C_1$, $q_D$ and interpret the former as an index of the drift of the process (valid for climate regime time scales $>\tau_c$ where by definition $H>0$), $C_1$ as an index of intermittency (spikiness), and
$q_D$ as a measure of the strength of the extremes, then we found that the beginning and end phases (1, 2, 8) have low drift, moderate intermittency, and strong extremes whereas the middle phases 3-7 have strong drift, weak intermittency, and weaker extremes (Fig. 12).

This paper is an early attempt to understand this unique very high resolution data set. In future work, we will extend our methodology to the EPICA paleo temperatures and to the scale by scale statistical relationship between the latter and the dust fluxes.

5    ## 5 Acknowledgements

SL's contribution to this fundamental research was unfunded and there were no conflicts of interest. FL acknowledges support by CONICYT projects Fondap 15110009 and Fondecyt 1171773, and the Millennium Nucleus Paleoclimate. MN Paleoclimate is a Millennium Nucleus supported by the Millennium Scientific Initiative of the Ministry of Economy, Development and Tourism (Chile)





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





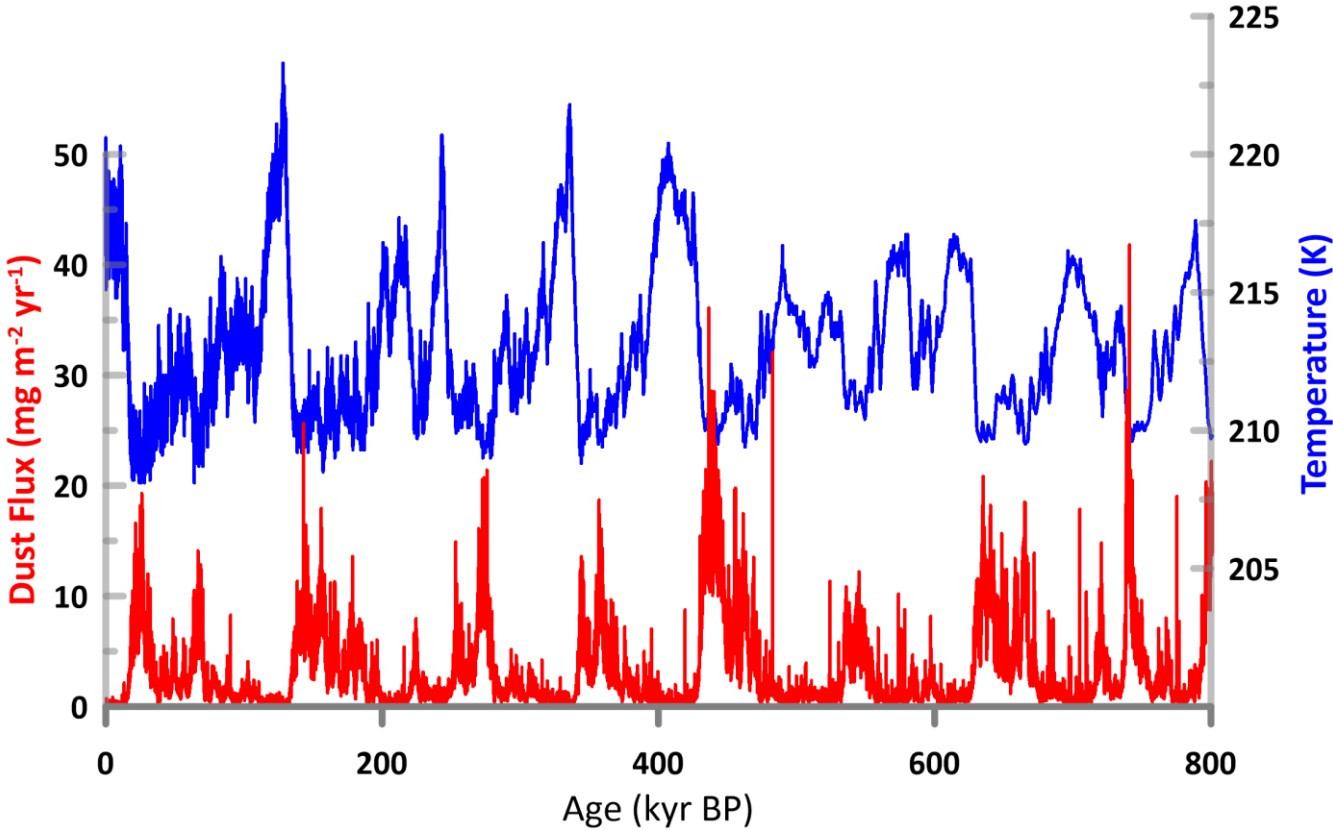

Figure 1: Temperature (blue) and dust flux (red) from the EPICA Dome C ice core (Jouzel et al., 2007; Lambert et al., 2012). The dust flux time series has 32,000 regularly spaced points (25 year resolution), the temperature series, has 5,752 points. The temperature data are irregularly spaced, and lose resolution as we go back into the past (number of temperature data points in successive ice ages: 3022, 1117, 521, 267, 199, 331, 134, 146). In both cases we can make out the glacial cycles but they are at best only quasi-periodic.





**Figure 2: A composite showing root mean square (RMS) Haar fluctuations (ΔT in units of °C) black, and RMS dust fluctuations analysed in this paper (red, in units of mg/m²/yr, (Lambert et al., 2012)). From left to right: thermistor temperatures at 0.0167s resolution (Lovejoy, 2018) , hourly temperatures from Landers Wyoming (Lovejoy, 2015) , daily temperatures from 75 °N (Lovejoy, 2015), EPICA Dome C temperatures (Jouzel et al., 2007), and two marine benthic stacks (Veizer et al., 1999; Zachos et al., 2001). The macroweather-climate transition is not in phase between the different records because the left ones (industrial side) are influenced by anthropogenic climate change, while the right data is pre-industrial natural variability. As elsewhere in this paper, the fluctuations were multiplied by the canonical calibration constant of 2 so that when the slopes are positive, the fluctuations are close to difference fluctuations. The various scaling regimes are indicated at the bottom. Adapted from (Lovejoy, 2017).**





**Figure 3a: Zooming out of the Holocene dust fluxes by octaves, by doubling the depth resolution from 1 cm (upper left) to 11m (lower right) resolution. Starting at the left and moving to the right and from top to bottom (see the ellipses on the first three in the sequence) we zoom out by factors of 2 in depth maintaining exactly 290 data points (effectively nondimensionalizing the depth; the small number of missing data points were not interpolated so that the final resolution is not exactly $2^{10}$cm = 10.24m). The temporal resolution is not exactly doubled due to the squashing of the ice column, the total duration (in years) of each section is indicated in each plot, the average temporal resolution of plots are: 0.24, 0.48, 0.98, 2.02, 4.32, 10.1 24.5, 54.1, 184, 434, 2710 yr. In order to fit all the curves on the same vertical scale, the dust fluxes were normalized by their mean over each segment. The means (in mg/m²/yr) are: 0.44, 0.38, 0.30, 0.36, 0.35, 0.33, 0.34, 0.39, 2.48, 2.18, 2.41 i.e. the first 8 plots have nearly the same vertical scales whereas the last three are about 6 times larger range. This means that all the plots except the last three are at nearly constant normalization.**





**Figure 3b: Same as Fig. 3a but for the absolute changes in dust flux normalized by the corresponding mean over the segment. The horizontal lines indicate the Gaussian probability levels for $p = 1/290$ (representing the mean extreme for a 290 point segment, red), as well as $p = 10^{-6}$ (orange). We see that each segment has several spikes that have Gaussian probabilities lower than $10^{-6}$.**





**Figure 4: Log-log plot of the Fourier spectrum of the (25yr)⁻¹ resolution dust concentration in frequency units of kyrs⁻¹ (red) and the same but of the logarithms of the flux (blue). Also shown is the average spectrum of the 5 year resolution data over the last 400 kyrs (green). There is a clear periodicity at about (100 kyrs)⁻¹. In the double power law fit (line plot), the transition frequencies are a little lower: $\omega_c$ = (160 kyr)⁻¹ (flux) and $\omega_c$ = (145 kyr)⁻¹ (log flux), although a Gaussian fit near the max gives a spike at (94±9 kyrs)⁻¹. Note that it is actually a little bit "wide" (two peaks) hence it is not perfectly periodic, and the amplitude is only about a factor 4 above the background. In comparison, the amplitude of the annual temperature frequency peak is several thousand times above the background (depending on the location) and is narrower (not shown).**

**The beta parameters are the exponents of the theoretical spectrum (see main text, the negative of the logarithmic slope) for the macroclimate (-2.5), climate (1.7), and macroweather (0.8) regimes.**

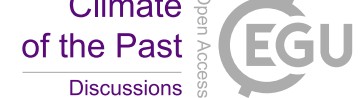





**Figure 5: Top set: successive segments of theoretical 100 kyr–long glacial cycles using usual (dimensional) time (present to past: bottom to top, the segment number is at the far right) with the 12.5 kyr phases indicated by vertical dashed lines. The short red lines indicate the interglacial dust minima. Each glacial-interglacial cycle is shifted by 25 units in the vertical for clarity. The red markers in the upper plot get mapped to the first dashed blue line in the lower plot.**

**Bottom set: successive cycles using nondimensional time (interglacial to interglacial) and then shifted by one phase to better line up with the usual time segments (the left most phase of the bottom line of the lower plot is zeroed). The average (nominal) resolution is 25 years. The interglacial dust minima were taken as 128.5, 243.5, 336, 407.5, 490, 614, 700, 789 kyrs B.P. and the data start at 373 yrs B.P. Each cycle is shifted by 25 units in the vertical for clarity. The data older than 789 kyrs were not used in these nondimensional cycles.**





**Figure 6: Top set: Averaging over the 8 cycles at 25 year resolution, we get the above picture: the mean is brown and the one standard deviation cycle to cycle variability is shown by the red. The dashed vertical lines give a further division into 8 x 12.5kyr segments, the 8 "phases" of the cycle.**

**Bottom set: the same but for the nondimensional time and shifted one phase to the right. The relative position of the interglacial at the first dashed line is indicated.**



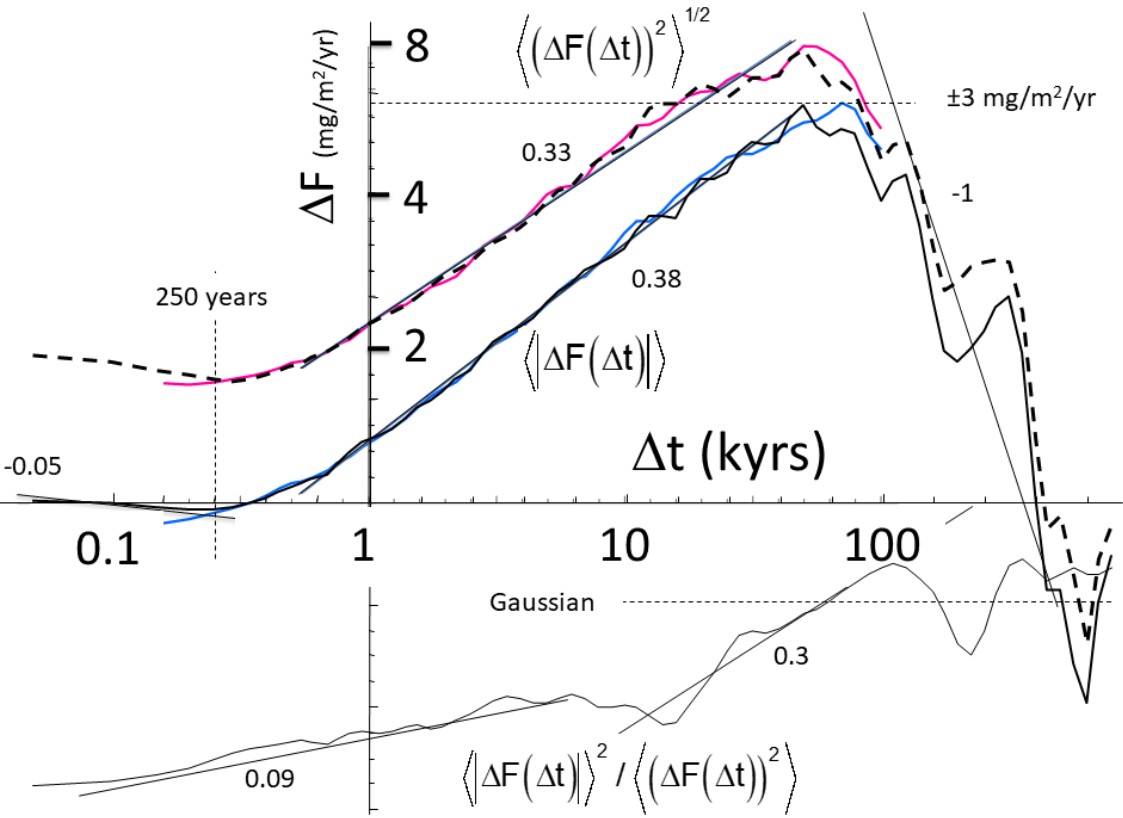

**Figure 7a: The Haar fluctuation analysis of the entire 800 kyr dust flux data set. Four curves are shown on this log-log plot. The dashed black and solid pink (top pair) represent RMS fluctuations for dimensional and non-dimensional time, respectively. The solid black and blue curves are the same but for the mean absolute (q =1) fluctuations. The curves with non-dimensional time lags have nominal (average) resolutions of 25 years and the fluctuation statistics are averaged over the 8 cycles. We can see that there is virtually no difference in the statistics in real time and in nondimensional time; the latter effectively stretches or compresses the time axis by ±19% which only implies a small additional fuzziness between the two.**

**Note that the peak in the curves occurs as expected at Δt ≈ 50kyrs i.e. at about a half cycle; and the horizontal dashed line shows that at this scale - corresponding to the largest difference in phases – the change in the mean absolute dust flux is about ± 3 mg/m²/yr.**

**Also shown (dashed vertical line) is the (average) time scale $\tau_c$ at which the transition from macroweather to climate occurs. Several reference lines (with the slopes/exponents indicated) are shown showing approximate scaling behaviours. One can see that as a general feature, the RMS and mean fluctuations tend to converge at large lags. This is quantified on the bottom curve which shows the square of the ratio of the mean to the RMS. If the statistics were Gaussian, then the ratio would be at the horizontal dashed line; we see some evidence for such behaviour at the very largest lags (≈> 80 kyrs, lower right where the ratio fluctuates about the Gaussian value; the statistics at these scales are poor).**





5    **Figure 7b: The q=1 structure function (S₁) for the most recent 400 kyrs at 5 year resolution (black) and S₁ for each of the 100 kyr segments (indicated by the numbers in kyrs). The figure shows the relatively small cycle to cycle variations in S₁ as well the high frequency H<0 regime more clearly than Fig. 7a (although the variability at the extreme left (Δt=10 years) is a little low probably due to instrumental smoothing of the 1cm data. A reference line slope *H* = 0.38 is also shown.**





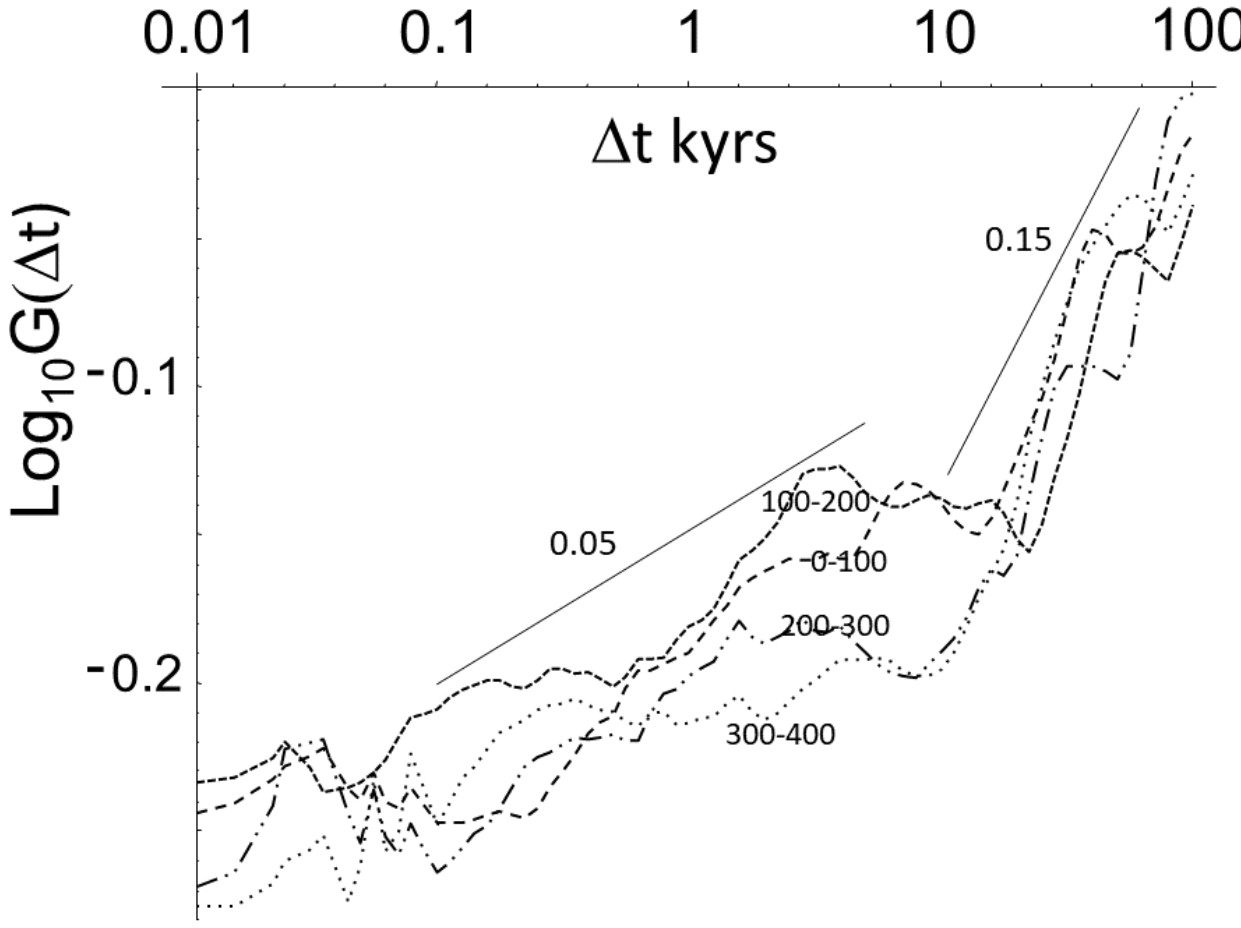

**Figure 7c: This shows the intermittency function G(Δt) whose exponent is the intermittency exponent $C_1$. The curves are from the first four successive 100 kyr segments of the 5 year resolution data in Fig. 7b. The intermittencies are compatible with those of the variance ratio (Fig. 7a) and are relatively stable from cycle to cycle. Reference lines with slopes $C_1 = 0.05$ and 0.15 are shown.**





**Figure 8: The probability distribution Pr(ΔF > s) of random changes in dust flux (ΔF) exceeding a fixed threshold s; in time at 25 year resolution (brown, 32,000 points), and in depth at 1cm resolution (black, 251,075 points corresponding to the last 400 kyrs). The straight lines indicate power law probability tails with exponents $q_D$ indicated. Also shown are the Gaussians with the same mean and standard deviations. In time, the maximum change in flux corresponds to about 28 standard deviations (i.e. to a Gauss probability ≈ $10^{-91}$), in depth, to 51 standard deviations (i.e. to p ≈ $10^{-455}$). As noted in the text, the tails are also much stronger than those of log-normal distributions.**







**Figure 9: The top row shows the intermittency function $G(\Delta t)$ (whose slope on the log-log plot is $C_1$) and the bottom row, the mean absolute Haar fluctuation $S_1(\Delta t)$ (whose slope on the log-log plot is $H$), the left column shows the result for each phase after averaging over the 8 cycles with the numbers next to each line indicate the phase number; the right hand column shows the result for each cycle after averaging over the phases. Whereas each cycle is fairly similar to every other cycle (the right column), each phase is quite different (the left column). We see the most significant difference is the fluctuation amplitude as a function of phase (lower left).**



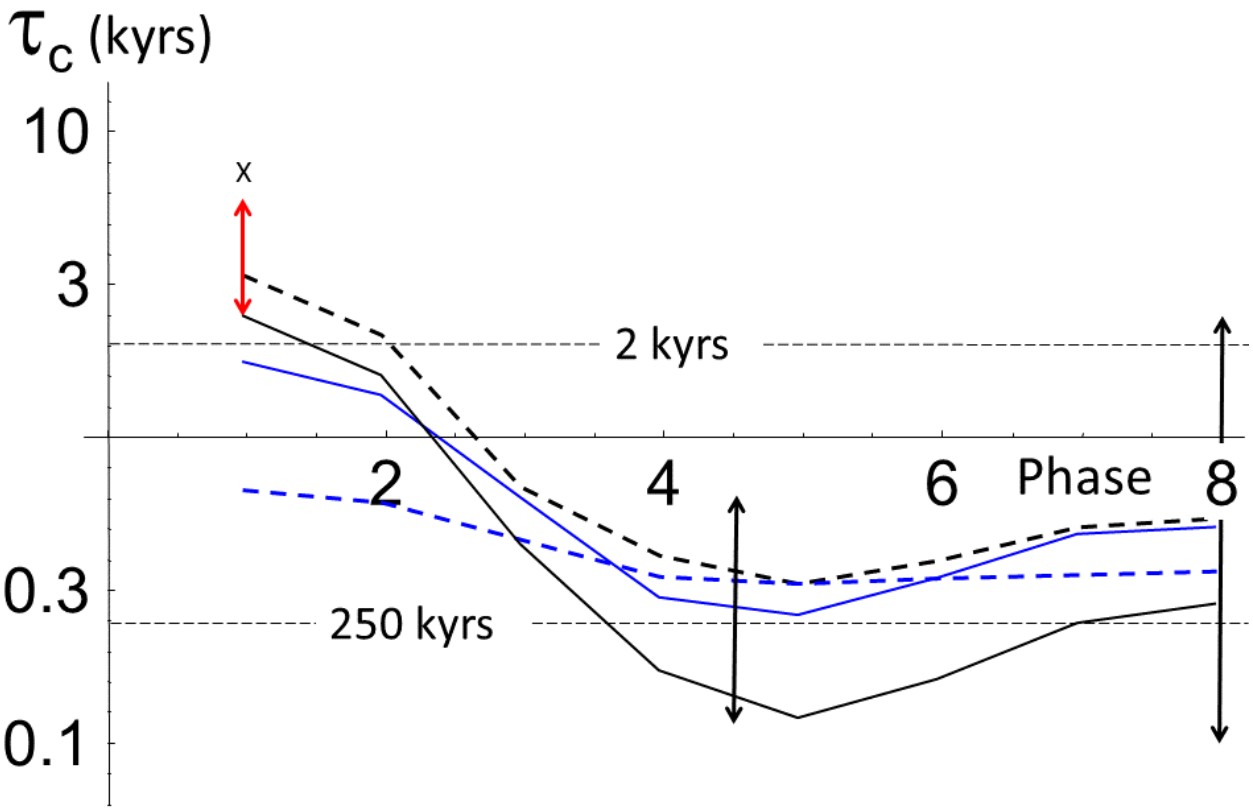

Figure 10: The transition scale $\tau_c$ estimated in two ways for each of the 8 phases and from two definitions of the phases. The first method (solid lines) used a bilinear fit to the (logarithm) of the Haar q=1 structure function (i.e. mean absolute fluctuation) as a function of log time lag $\Delta t$. To obtain robust results, a small $\Delta t$ region with the slope -0.05 and a large $\Delta t$ slope +0.25 was imposed

5    with the transition point ($\tau_c$) determined by regression. This was done for each segment and cycle. For each phase there were thus 8 transition scales, which were used to calculate the mean of the logarithm of $\tau_c$ and its standard deviation. Results are shown for dimensional (segments, blue) and nondimensional time (cycles, black).

      The second method used to estimate $\tau_c$ was graphical and relied on a somewhat subjective fitting of scaling regimes and transitions, but without imposing small and large $\Delta t$ slopes (exponents *H*). The results are shown in dashed lines, they are quite

10    similar although we can note some differences for the first phase (dimensional, blue) and the middle phases (nondimensional, black). There is also considerable cycle to cycle spread that was quantified by the standard deviations. In order to avoid clutter, typical spreads are shown by the double headed black arrows. Dashed horizontal lines show the ensemble mean transition scale (about 250 years) as well as ensemble mean for phases 1 and 2 (around 2 kyrs), which stands out compared to the rest of the phases. The red arrow shows one standard deviation for the nondimensional first phases, while the X marks the value of the

15    Holocene $\tau_c$ (7.9 kyr) just outside the 1-sigma limit. Thus the Holocene was a bit extreme, but not an outlier.



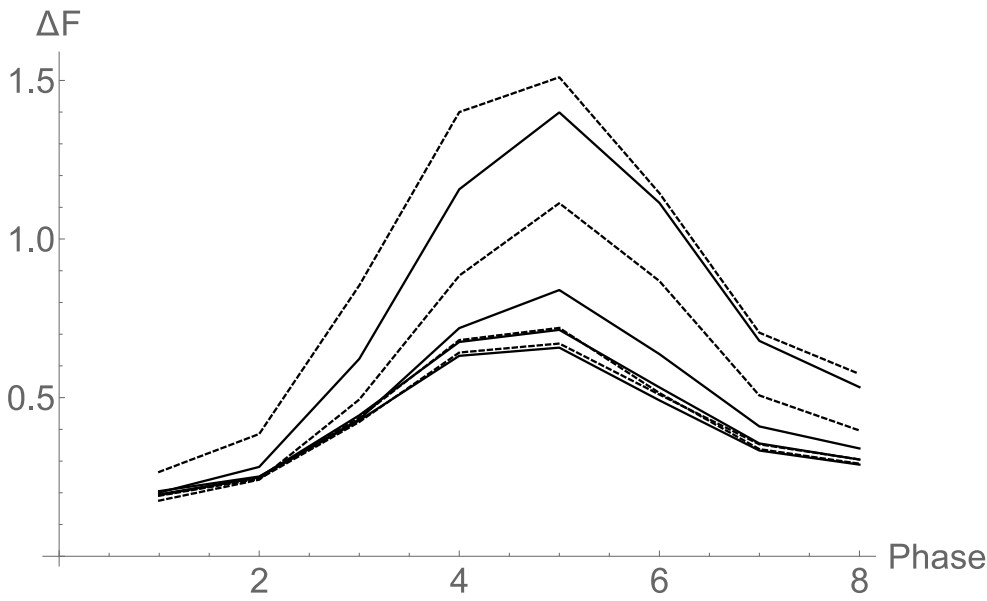

5      **Figure 11:**    **Using nondimensional time, the Haar fluctuations for Δt = 50, 100, 200, 400, 800, 1600, 3500, 7000 years averaged over all the cycles is shown (bottom to top, alternating solid and dashed), dimensions mg/m²/yr.    The cycle to cycle variability is about a factor of 2. We see that the variability is much larger in the middle phases of the cycle than at the beginning and end phases and that it increases more rapidly with scale (larger H) in the middle phases.**



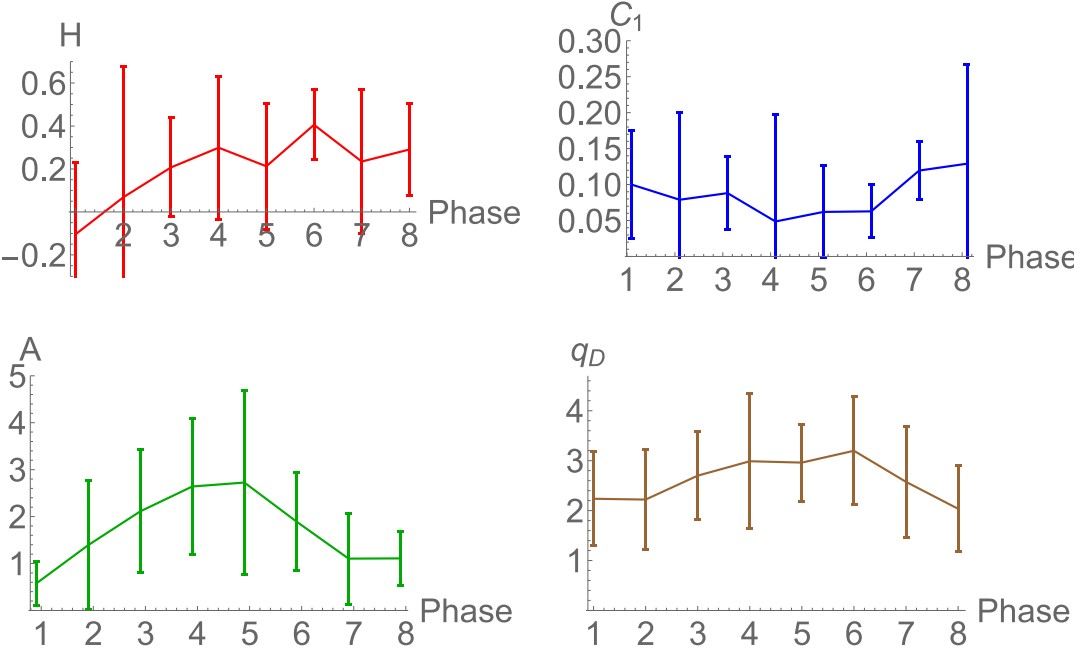

**Fig.12:** **The fluctuation and intermittency exponents H and $C_1$ (top row), and fluctuation amplitude A and extremeness $q_D$ (bottom row) are estimated over the range $500 - 3000$ years, as a function phase with the standard deviations from the cycle to cycle variability (all using nondimensional time). The upper left (H) plot shows that the cycles start with low drift but become driftier in the middle and later phases. The intermittency ($C_1$, upper right) is moderate at the beginning and end of the cycles, and a little weaker in the middle. The lower left shows the amplitude of the fluctuations at 25 years determined by the standard deviation of the dust flux (units: mg/m$^2$/yr). We see that the flux has low amplitude fluctuations at the beginning and end of the cycles and 3-4 times higher amplitude fluctuations in the middle. The lower right shows the probability exponent $q_D$ estimated from the 25 year resolution data for each phase; the extreme 5% of the flux changes were used to determine the exponent in each phase; the cycle to cycle spread is indicated by the error bars (overall average over the phases: $q_D = 2.62\pm0.42$). This qualitative interpretation is in accord with the data in Fig. 5.**