# Peer review of "High resolution EPICA ice core dust fluxes: intermittency, extremes and Holocene stability"

_Climate of the Past, 2018_

## Referee Comment (RC1) · Anonymous Referee #1 · 24 Sep 2018

This paper presents statistical analyses of a dust record from the EPICA record to describe the temporal variability of the last 800 kyr.

General comments

I do not think that the analyses reported by the paper contain errors, although I did not try to replicate the results. It is difficult to make out the goal of the paper, which appears as a sequence of statistical analyses that the first author seems to have repeated in quite a few recent papers (listed in the manuscript and others). I am surprised that the authors do not cite the paper of Huybers and Curry (Nature, Links between annual, Milankovitch and continuum temperature variability, 2006) that already discussed such

statistical analyses, albeit on other datasets. So, my feeling is that there is very little new understanding in the manuscript.

Specific comments

The introduction should state clearly the scientific question that will be tackled in the manuscript (not just list scientific problems), and the conclusion should state how the obtained results served to solve the problem (or not). Without such a reorganization, it is very difficult to assess the importance of the paper. At present, the conclusion essentially paraphrases the results, and seems to depend on choices of parameters (like time interval discretization). The authors did something that they are the only ones to understand, to reach a conclusion that is very hard to exploit.

I would appreciate that the methodologies used in the manuscript appear under a "Methodology" section. Most of the equations (so, what is done, or not) appear in the "Results" section. This makes the separation of what is new from what is supposed to be known rather tedious.

I see no assessment of uncertainty (on data or ice core dating) in the manuscript. Would that mean that the results would be insensitive to the chronology?

The authors use a dust flux reconstruction (I assume computed from a dust content, divided by time increments). I am not extremely familiar with such Antarctic records, but a rapid bibliographic search reveals that similar data (dust, chemical species) in Greenland ice cores show that the logarithm of dust (and chemical species) are heavily correlated to isotopic data (Yiou, R., Fuhrer, K., Meeker, L. D., Jouzel, J., Johnsen, S., & Mayewski, P. A. (1997). Paleoclimatic variability inferred from the spectral analysis of Greenland and Antarctic ice‐core data. Journal of Geophysical Research: Oceans, 102(C12), 26441-26454; Mayewski, P. A., Meeker, L. D., Twickler, M. S., Whitlow, S., Yang, Q., Lyons, W. B., & Prentice, M. (1997). Major features and forcing of high‐latitude northern hemisphere atmospheric circulation using a 110,000‐year‐long glaciochemical series. Journal of Geophysical Research: Oceans, 102(C12), 26345-

26366; Fuhrer, K., Wolff, E. W., & Johnsen, S. J. (1999). Timescales for dust variability in the Greenland Ice Core Project (GRIP) ice core in the last 100,000 years. Journal of Geophysical Research: Atmospheres, 104(D24), 31043-31052.). So, why not consider the logarithm of dust flux?

The authors repeat several times that dust flux is not Gaussian. This is rather trivial, given that it always have positive values. Why should fitting a Gaussian process to dust be a reasonable null hypothesis to reject? Dust flux is generally modeled by a transport equation, the solutions of which are like multiplicative noise.

The climate interpretation of the results obtained by the deployment of this arsenal also seems to be a problem. I expect that such an interpretation is natural when considering a variable of the system. The authors use an observable (dust flux) that might be obtained by a complex transformation of a driving variable of the climate system (like temperature, or temperature gradients, or pressure variations). To what extent what is learnt from the analyses does not just reflect something on the complex transformation of an underlying driving variable, rather than the dynamics of the variable itself? Since the authors never discuss the physical meaning of the data they analyze and climate variations (or only in a superficial way), I could doubt that any physical interpretation can be deduced from the analyses.

Minor comments

I have too many comments on the manuscript. I will limit them to 10.

p. 1, l. 10: the first sentence of the abstract does not make any sense to me (see the paper of Huybers & Curry, 2006, and many others).

p. 1, l. 19: at this point I would need to know what foreground and background processes are.

p. 2, l. 1-8: all this seems to be a personal opinion.

What is the use of Figure 2? What does "fluctuation tend to persist because they are

unstable" mean? Where is the peak in Fig. 2? What is $\tau_c$? It is never defined. I do not think that Petit et al. computed any $\tau_c$. Their statement on agriculture was a perspective, not a result of the paper itself!

p. 3, l. 2: The notion of stability/instability was different in the papers of Petit et al. and Berner et al. The paper of Berner et al. discusses Subpolar North Atlantic ocean dynamics instabilities, which are not accessible by Antarctic ice cores.

p. 3, l. 5: Antarctic dust content is necessarily connected to the atmospheric circulation. Dust records in Arctic ice core reflect the atmospheric circulation. Why not compare dust records of both hemispheres? What is the precise question that the authors want to address?

p. 4, l. 26: What is a statistical symmetry?

p. 8, l. 18: I do not understand this equation. If s>1, then any positive power of s gives a number that is larger than one. How can a probability be larger than one?

p. 9, l. 5: What do the authors mean by "extremes much occur too frequently"? How is this related to the analysis of the paper? The paragraphs between l 3-13 are incomprehensible for someone else than the authors.

p. 9, l.14-onward: In Taleb's book, black swans do not necessarily refer to Lévy flights, but to events whose features cannot be anticipated (like the war in Lebanon, the success of Harry Potter, etc.). Gray swans (Taleb's spelling) are those events for which some sort of anticipation can be provided. Incidentally, Taleb also writes on confirmation bias, which is one of the flaws that I tried to outline when the authors interpret their analyses (e.g. p. 9, l. 30). Therefore, the overall understanding of the cited literature could be improved.
* * *

---

## Referee Comment (RC2) · Anonymous Referee #2 · 5 Nov 2018

**SUMMARY**
Lovejoy & Lambert describe and analyze a high resolution dust flux time series from the EPICA Dome C ice core using 'fluctuation analysis' and spectral analysis, both over the full 800,000 year-record as well as for individual glacial cycles.

**GENERAL COMMENTS**

1. The authors apply interesting and novel concepts to a high-resolution record, so there should be a considerable potential for new insights.

2. However, the structure of the paper is too chaotic (almost intermittent) with data, methods, results and discussion randomly mixed. This renders the result barely readable and obscures potential interesting results. The introduction and discussion is currently myopic.

3. There are inconsistencies in the results, with the Holocene transition time Tau_C identified at 4, 3-5 and 7.9 kyrs at different points in the paper.

Without a substantial restructuring this paper is not acceptable.

**DETAILED COMMENTS**

**p1l12/13** please indicate that you are using the definitions you provided in earlier work. These are not common concepts in (palaeo)climatology.

**p1l20** what are the hypotheses underlying these two analyses with fixed/variable cycle durations?

**p1l25** $\tau_c$=4kyrs

**p1l30–p2l4** A sharp peak in a spectrum due to a periodic component in the signal would be blurred and broadened by temporal uncertainty – which is, conceivably, larger in the earlier parts of the Pleistocene, the "41kyr world", for which data is based off marine records. The ratio of age uncertainty to period length is less favourable then, and many records are orbitally tuned (although possibly not using the analyzed signal).

**p2l7** A brief definition of macroweather vs. climate regimes would be helpful here.

**p2l16020** Mitchell's drawn spectrum was conceptual, and we know that it isn't accurate from earlier work (Huybers & Curry, 2006; Laepple & Huybers 2013)

**p2l34/p3l1** This lacks recent literature. Interglacial vs. Glacial period climate scaling and variability have been repeatedly compared in the literature. Whereas Ditlevsen et al. (1996) and Shao & Ditlevsen (2016) investigated the scaling properties for the different climate periods and found strong differences, Rehfeld et al (2018) suggested that on millennial scale Glacial vs Holocene variability is approximately 4:1.

**p3l4-9** This implies that you are analyzing dust as a temperature proxy, but the two signals scale differently. Dust concentrations are non-negative and non-Gaussian by definition.

**p4l1** How were these spectral analyses performed? Why are there no confidence intervals? It would help if fluctuation analysis and spectral analyses were performed and displayed for the same datasets, given that most readers would be familiar with the latter.

**p4l14/15** Presumably these estimates (like most others in this paper) have some uncertainty. Please state them!

**p5l22** : Definitions belong to the methods, not the results. It would benefit the paper – and justify it – if methods, results and discussion were separated. Then the authors could devote a couple of paragraphs to the actual discussion of the processes and dynamics suggested by their results - such as the progression of deserts during Glacials that could be one of the reasons for the larger variance mid-cycle - which are lacking.

**p6l26** To bear in mind: Mitchell draws a spectrum for temperature (conceptually), but data-based estimates have to rely on proxies for temperature, which potentially nonlinearly transforms the original processes.

**p6l4** Haar fluctuations and intermittency should be introduced in a methods section.

**p7l6/7** Please add a statistical test, considering age uncertainty, uncertainty in the transfer function and measurement noise. Otherwise robustness of the results cannot be judged

**p9** Dust concentrations cannot be Gaussian, as they are counted variables and by definition positive definite.

**p10l18** Holocene $\tau_c$=7.9kyrs

**p12l22** Holocene $\tau_c$=3-5kyrs

**p13l1-3** Maybe this early analysis can be progressed to an actual robust analysis of this dataset.

**Dataset** Is this the dataset used? Please provide links to the versions and/or where the data is available. https://doi.pangaea.de/10.1594/PANGAEA.779311

**Figure 2** presumably the hourly temperatures from Landers Wyoming and the daily temperatures at 75 degrees North were not measured by the authors, could you give the original references, please?

**Figure 3b** The axes here are unreadable. Rather than show the obvious (Gaussian assumption makes no sense for dust fluxes), why not consider nonparametric confidence levels, or show at least the log of the dust flux).

**Figure 4** The decrease in power towards the lowest frequencies (>400,000, beta=-2) may well be an artifact: By construction, periods longer than the time-series length divided by two cannot be interpreted, and rules of thumb/good practice is to stick to 1/3rd of the record length. For this 800,000 year-record this would mean that the spectrum could be considered estimable up to timescales of ∼1/266,000 years. It would further be good practice to subtract at least a linear trend (Chatfield 2016).

**Figure 10** How can the Holocene, being 11,700 years long, have a transition time scale $\tau_c$ of 7,900 years?

**REFERENCES**

Lambert, F., Bigler, M., Steffensen, J. P., Hutterli, M., & Fischer, H. (2012). Centennial mineral dust variability in high-resolution ice core data from Dome C, Antarctica. Climate of the Past, 8(2), 609-623.

Huybers, P., & Curry, W. (2006). Links between annual, Milankovitch and continuum temperature variability. Nature, 441(7091), 329.

Laepple, T., & Huybers, P. (2013). Reconciling discrepancies between Uk37 and Mg/Ca reconstructions of Holocene marine temperature variability. Earth and Planetary Science Letters, 375, 418-429.

Fredriksen, H. B., & Rypdal, K. (2016). Spectral characteristics of instrumental and climate model surface temperatures. Journal of Climate, 29(4), 1253-1268.

Ditlevsen, P. D., Svensmark, H., & Johnsen, S. (1996). Contrasting atmospheric and climate dynamics of the last-glacial and Holocene periods. Nature, 379(6568), 810.

Rehfeld, K., Münch, T., Ho, S. L., & Laepple, T. (2018). Global patterns of declining temperature variability from the Last Glacial Maximum to the Holocene. Nature, 554(7692), 356.

Shao, Z. G., & Ditlevsen, P. D. (2016). Contrasting scaling properties of interglacial and glacial climates. Nature communications, 7, 10951.

Chatfield, C. (2016). The analysis of time series: an introduction. CRC press

---

## Author Comment (AC1) · 1 Dec 2018

This paper presents statistical analyses of a dust record from the EPICA record to describe the temporal variability of the last 800 kyr.

General comments I do not think that the analyses reported by the paper contain errors, although I did not try to replicate the results. It is difficult to make out the goal of the paper, which appears as a sequence of statistical analyses that the first author seems to have repeated in quite a few recent papers (listed in the manuscript and others). I am surprised that the authors do not cite the paper of Huybers and Curry (Nature, Links between annual, Milankovitch and continuum temperature variability, 2006) that

already discussed such statistical analyses, albeit on other datasets.

Au: That paper is now mentioned several times. However, our paper goes well beyond the Huybers paper both at the level of data quality and quantity as well as the types of statistical characterization that were made. These support a new phase by phase understanding of the scale by scale variability including intermittency of the last 8 glacial cycles.

So, my feeling is that there is very little new understanding in the manuscript.

Au: Our paper was indeed largely empirical, providing original characterizations of a unique climate data set. But it is unfair to condemn the paper as providing little understanding: understanding must be based on the highest quality data and on the most systematic, highest quality analysis of that data. Thanks to our analysis new understandings must accommodate the following observations: a) That the commonly used log transformation of dust fluxes does not significantly change the variability – whether it be the spectrum, or the nature of the extremes. Since the log dust flux is not a physically significant variable, we conclude that the transformation is not necessarily helpful beyond a visual aid in plots. b) To our knowledge this is the first time that detailed comparisons between all of the last 8 glacial cycles has been made. We showed in detail that the cycles are statistically quite close to each other while being systematically different according to the phase of the cycle. c) This is the first quantitative characterization of climate stability as a function of cycle and phase. It allows for a clarification idea of Holocene exceptionalism – at least with respect to this high latitude data set.

Specific comments The introduction should state clearly the scientific question that will be tackled in the manuscript (not just list scientific problems), and the conclusion should state how the obtained results served to solve the problem (or not). Without such a reorganization, it is very difficult to assess the importance of the paper. At present, the conclusion essentially paraphrases the results, and seems to depend on choices of parameters (like time interval discretization). The authors did something

that they are the only ones to understand, to reach a conclusion that is very hard to exploit. I would appreciate that the methodologies used in the manuscript appear under a "Methodology" section. Most of the equations (so, what is done, or not) appear in the "Results" section. This makes the separation of what is new from what is supposed to be known rather tedious.

Au: We have re-organized the paper as suggested.

I see no assessment of uncertainty (on data or ice core dating) in the manuscript. Would that mean that the results would be insensitive to the chronology?

Au: The uncertainty of our results is actually discussed at large. In addition to uncertainty estimates of statistical variables, we performed analyses both in time and in depth, thus providing one analysis that is indeed insensitive to chronology. In addition, we compare two different chronologies: one in time (based on the official dating) and one using nondimensional time; the fraction of a glacial cycle. The manuscript discusses the fact that our results are fairly robust with respect to this quite significant change. In the new iteration, we tried to make this more conspicuous.

The authors use a dust flux reconstruction (I assume computed from a dust content, divided by time increments).

Au: In essence, yes. The flux is calculated by multiplying the dust particle concentration with the accumulation rate.

I am not extremely familiar with such Antarctic records, but a rapid bibliographic search reveals that similar data (dust, chemical species) in Greenland ice cores show that the logarithm of dust (and chemical species) are heavily correlated to isotopic data (Yiou, R., Fuhrer, K., Meeker, L. D., Jouzel, J., Johnsen, S., & Mayewski, P. A. (1997). Paleoclimatic variability inferred from the spectral analysis of Greenland and Antarctic ice core data. Journal of Geophysical Research: Oceans, 102(C12), 26441-26454; Mayewski, P. A., Meeker, L. D., Twickler, M. S., Whitlow, S., Yang, Q., Lyons, W. B., &

Prentice, M. (1997). Major features and forcing of high latitude northern hemisphere atmospheric circulation using a 110,000 year long glaciochemical series. Journal of Geophysical Research: Oceans, 102(C12), 26345-26366; Fuhrer, K., Wolff, E. W., & Johnsen, S. J. (1999). Timescales for dust variability in the Greenland Ice Core Project (GRIP) ice core in the last 100,000 years. Journal of Geophysical Research: Atmospheres, 104(D24), 31043-31052.).

Au: The correlation between dust flux and isotopic series depends greatly on the time scale at which the correlation is determined. A systematic demonstration of this fact will be given in a new paper in preparation; it is beyond the scope of the present paper.

So, why not consider the logarithm of dust flux?

Au: We did consider dust flux but this was apparently missed by the referee. To make the point more strongly, we added a new plot of the probability distribution of the logarithms of the flux, further confirming that the transformation only results in minor changes in the statistics. We also added a new spike plot of the log transformed data showing that the qualitative nature of the dust variability is not changed. The spectrum, intermittency and extremes are nearly unaffected. We cannot avoid dealing with the extreme spikiness of the dust flux. Fortunately, unlike standard approaches, our methods are ideally suited for such spiky analyses.

The authors repeat several times that dust flux is not Gaussian. This is rather trivial, given that it always have positive values.

Au: Yes, except that we were discussing the increments of the process being non-Gaussian and this is less trivial. In addition we showed that the log transformation does not render the process Gaussian either.

Why should fitting a Gaussian process to dust be a reasonable null hypothesis to reject? Dust flux is generally modeled by a transport equation, the solutions of which are like multiplicative noise.

Au: Yes, exactly! Our scaling methodology is based on an understanding of multiplicative turbulent cascades that was developed over the last 35 years (multifractals). The generic result is multiscaling with power law extremes, exactly as found here. We further underscore this point in the new iteration.

The climate interpretation of the results obtained by the deployment of this arsenal also seems to be a problem. I expect that such an interpretation is natural when considering a variable of the system. The authors use an observable (dust flux) that might be obtained by a complex transformation of a driving variable of the climate system (like temperature, or temperature gradients, or pressure variations). To what extent what is learnt from the analyses does not just reflect something on the complex transformation of an underlying driving variable, rather than the dynamics of the variable itself? Since the authors never discuss the physical meaning of the data they analyze and climate variations (or only in a superficial way), I could doubt that any physical interpretation can be deduced from the analyses.

Au: We saw that a rather drastic transformation of the flux - a log transformation – does not fundamentally alter its extreme statistical characteristics. This is perhaps not surprising because there are wide scale ranges over which the fluxes are scale invariant and one expects that this reflects an underlying symmetry of the dynamics (no matter what the relation of the latter is to the fluxes). Similarly, the stability/instability conclusions are likely to be robust so that our analyses therefore give a robust indicators of scaling and the limits of the scaling regimes, and this independently of the link between dust fluxes and more conventional climate parameters. But we agree that the climatic interpretation was hidden within the technical jargon. We tried to clarify this aspect in the revised version.

Minor comments I have too many comments on the manuscript. I will limit them to 10.

p. 1, l. 10: the first sentence of the abstract does not make any sense to me (see the paper of Huybers & Curry, 2006, and many others).

Au: We removed the sentence.

p. 1, l. 19: at this point I would need to know what foreground and background processes are.

Au: We removed this reference.

p. 2, l. 1-8: all this seems to be a personal opinion. What is the use of Figure 2?

Au: It situates the dust variability in the larger scheme of temperature and temperature proxy variability. It shows that it is quantitatively quite similar.

What does "fluctuation tend to persist because they are unstable" mean? Where is the peak in Fig. 2?

What is ïĄťc ? It is never defined.

Au: ïĄťc is the time scale at which the scaling regime switches from the "macroweather" to the "climate" regime. It was defined in line 24, p1. The definition was repeated on line 33 of p2 and on line 8 of p8 and on line 3 of p10 and on line 8 of p12. We repeated the definition many times because of its importance to the paper. We do not understand how it was missed.

I do not think that Petit et al. computed any ïĄťc. Their statement on agriculture was a perspective, not a result of the paper itself!

Au: Exactly our point! They "eyeballed" their series whereas we have an objective quantification. Nevertheless we removed this sentence.

p. 3, l. 2: The notion of stability/instability was different in the papers of Petit et al. and Berner et al. The paper of Berner et al. discusses Subpolar North Atlantic ocean dynamics instabilities, which are not accessible by Antarctic ice cores.

Au: Our point was that in both cases, conclusions were drawn with implications about Holocene stability. But we removed that paragraph.

p. 3, l. 5: Antarctic dust content is necessarily connected to the atmospheric circulation. Dust records in Arctic ice core reflect the atmospheric circulation. Why not compare dust records of both hemispheres? What is the precise question that the authors want to address?

Au: The comparison between hemispheres is beyond the scope of this paper. We attempt to clarify the nature of the glacial cycles and quantify their variability as functions of the phases of the cycle.

p. 4, l. 26: What is a statistical symmetry?

Au: It is an invariance of some aspect of the statistics under some transformation. Here, an invariance of statistical exponents under time dilations and contractions (temporal scale invariance). This is now mentioned in the new iteration.

p. 8, l. 18: I do not understand this equation. If s>1, then any positive power of s gives a number that is larger than one. How can a probability be larger than one?

Au: There was a typo in the exponent, there should have been a minus sign before the qD. We apologize. This has been fixed.

p. 9, l. 5: What do the authors mean by "extremes much occur too frequently"? How is this related to the analysis of the paper? The paragraphs between l 3-13 are incomprehensible for someone else than the authors.

Au: We meant to say that "extremes occur much too frequently for the process to be Gaussian". This conclusion implies that nonstandard theoretical frameworks are required for understanding dust fluxes. We propose multifractal scaling as such a framework.

p. 9, l.14-onward: In Taleb's book, black swans do not necessarily refer to Lévy flights, but to events whose features cannot be anticipated (like the war in Lebanon, the success of Harry Potter, etc.). Gray swans (Taleb's spelling) are those events for which some sort of anticipation can be provided. Incidentally, Taleb also writes on confirmation bias, which is one of the flaws that I tried to outline when the authors interpret their analyses (e.g. p. 9, l. 30). Therefore, the overall understanding of the cited literature could be improved.

Au: Yes, we know and tried to make this distinction in the text. Yet, the term "black swan" is often used in place of the rarely used "grey swan" expression, so we used "black swan" even if it is not quite the original usage. We removed this passage to avoid confusion.

Please also note the supplement to this comment:
https://www.clim-past-discuss.net/cp-2018-110/cp-2018-110-AC1-supplement.pdf

---

## Author Comment (AC2) · 1 Dec 2018

GENERAL COMMENTS

1. The authors apply interesting and novel concepts to a high-resolution record, so there should be a considerable potential for new insights.

Au: Thank you for your appreciation.

2. However, the structure of the paper is too chaotic (almost intermittent) with data, methods, results and discussion randomly mixed. This renders the result barely readable and obscures potential interesting results. The introduction and discussion is

currently myopic.

Au: We agree and have restructured the paper and rewritten various sections.

3. There are inconsistencies in the results, with the Holocene transition time Tau_C identified at 4, 3-5 and 7.9 kyrs at different points in the paper. Without a substantial restructuring this paper is not acceptable.

Au: The problem is that there were 8 phases and 8 cycles yielding 64 different ïĄťc values. Each was estimated in three different ways (two different chronologies and two different estimation techniques). Various averages over phases and cycles were also given. We have rewritten some sections to make it more consistent.

DETAILED COMMENTS p1l12/13 please indicate that you are using the definitions you provided in earlier work. These are not common concepts in (palaeo)climatology.

Au: Done.

p1l20 what are the hypotheses underlying these two analyses with fixed/variable cycle durations?

Au: These are described in the text. The segment definition is based on the spectrum with peak near 100kyrs, the cycle definition is based on the glacial maxima.

p1l25 _c=4kyrs

Au: ïĄťc is not exactly 4kyrs which is why we use the $\approx$ sign.

p1l30–p2l4 A sharp peak in a spectrum due to a periodic component in the signal would be blurred and broadened by temporal uncertainty – which is, conceivably, larger in the earlier parts of the Pleistocene, the "41kyr world", for which data is based off marine records. The ratio of age uncertainty to period length is less favourable then, and many records are orbitally tuned (although possibly not using the analyzed signal).

Au: The Huybers [Huybers, 2007] reconstruction avoids orbital tuning yet shows a

strong spectral peak at (41kyrs)-1, but only in the period before 800kyrs. We have removed this section as it was not absolutely relevant to the scope of the paper.

p2l7 A brief definition of macroweather vs. climate regimes would be helpful here.

Au: Done.

p2l16020 Mitchell's drawn spectrum was conceptual, and we know that it isn't accurate from earlier work (Huybers & Curry, 2006; Laepple & Huybers 2013).

Au: We agree. The point is precisely that because the inaccuracy was quantitatively astronomical, that Mitchell's conceptual framework is untenable. We have removed the historic context to simplify the paper.

p2l34/p3l1 This lacks recent literature. Interglacial vs. Glacial period climate scaling and variability have been repeatedly compared in the literature. Whereas Ditlevsen et al. (1996) and Shao & Ditlevsen (2016) investigated the scaling properties for the different climate periods and found strong differences, Rehfeld et al (2018) suggested that on millennial scale Glacial vs Holocene variability is approximately 4:1.

Au: Thank you. We have added a paragraph with these references.

p3l4-9 This implies that you are analyzing dust as a temperature proxy, but the two signals scale differently. Dust concentrations are non-negative and non-Gaussian by definition.

Au: The dust and temperature have scale dependent links; we will investigate this in detail in a future publication, but it is not true that dust is a temperature proxy. However, scaling regimes and the transition scale between a stable and unstable regimes are fairly fundamental characteristics and should be observable in either record.

p4l1 How were these spectral analyses performed? Why are there no confidence intervals?

Au: The spectra were analyzed using FFT with standard Hanning windows with the

smooth (red and blue) curves obtained by averaging over logarithmically spaced bins as indicated in the text. For the Green curve, we have followed the usual practice in turbulence which is to display the full spectrum. In this case, the uncertainty is directly judged by the spread around the mean.

It would help if fluctuation analysis and spectral analyses were performed and displayed for the same datasets, given that most readers would be familiar with the latter.

Au: The fluctuation analyses of the same data were indeed shown in fig. 7 (Figure 5 in the new version).

p4l14/15 Presumably these estimates (like most others in this paper) have some uncertainty. Please state them!

Au: As usual, the uncertainty depends on your basic model. Here, for the fraction in the background, various different models were compared so that the reader can judge. Later, the difficulty is that the only way to satisfactorily judge the parameter uncertainties is to have a well defined stochastic model of the process. That being said, standard regression uncertainties could be used as uncertainty proxies for many of the exponents. Unfortunately the real source of uncertainty is not the traditional statistical variation about a regression fit (e.g. the slope of a straight line on a log-log plot), it is rather the difficulty of objectively choosing the scale range over which the regression exponents are estimated. For example, in one of the methods for estimating ïĄťc, an objective method for determining the transition between two scaling regimes was used. But this only works if the high and low frequency exponents are fixed beforehand; a kind of "bootstrap". In the end, we estimated the uncertainty by comparing three different methods (fig. 10) and by comparing estimates from different cycles (fig. 12). All told a huge effort was made to quantify uncertainty.

p5l22 : Definitions belong to the methods, not the results. It would benefit the paper – and justify it – if methods, results and discussion were separated. Then the authors could devote a couple of paragraphs to the actual discussion of the processes and dy-

namics suggested by their results - such as the progression of deserts during Glacials that could be one of the reasons for the larger variance mid-cycle - which are lacking.

Au: We agree. We have re-organized the paper to separate better methods, results, discussion.

p6l26 To bear in mind: Mitchell draws a spectrum for temperature (conceptually), but data-based estimates have to rely on proxies for temperature, which potentially nonlinearly transforms the original processes.

Au: Yes, But Mitchell's spectrum still implied that successive million year temperature averages differed by microKelvins. Whereas the usual paleo calibrations imply differences of about 1 Kelvin. In our opinion, the latter is more plausible.

p6l4 Haar fluctuations and intermittency should be introduced in a methods section.

Au: Done.

p7l6/7 Please add a statistical test, considering age uncertainty, uncertainty in the transfer function and measurement noise. Otherwise robustness of the results cannot be judged.

Au: Again, standard statistical tests require a stochastic model of the process. We are attempting a kind of a bootstrap whereby a statistical characterization of the data is given and uncertainties judged by a series of measures including cycle to cycle comparisons. The aim is indeed to develop a stochastic model that could reproduce the dust series as a single realization of stochastic process with well defined parameters. In that case, the uncertainties are measured with respect to the model. We are not quite there yet and so have to judge the uncertainty by a series of less precise alternatives, which give us a good idea of the uncertainty range, though.

p9 Dust concentrations cannot be Gaussian, as they are counted variables and by definition positive definite.

Au: Actually the claim was that the process might be considered Gaussian - meaning that the increments (denoted here by ïĄĎF) - might be Gaussian or approximately Gaussian variables. Although it is indeed a terrible model for dust, it cannot be trivially rejected simply on the basis of the observation that the signal itself is positive definite over a range.

p10l18 Holocene _c=7.9kyrs.

Au: Thanks.

p12l22 Holocene _c=3-5kyrs.

Au: Thanks.

p13l1-3 Maybe this early analysis can be progressed to an actual robust analysis of this dataset. Dataset Is this the dataset used? Please provide links to the versions and/or where the data is available. https://doi.pangaea.de/10.1594/PANGAEA.779311

Au: Yes, we also hope that this is the beginning of new approaches to analyzing dust fluxes.

Figure 2 presumably the hourly temperatures from Landers Wyoming and the daily temperatures at 75 degrees North were not measured by the authors, could you give the original references, please?

Au: The Landers series was from data from the US climatalogical data network and the 75oN data were from the Twentieth Century Reanalysis. Full references and discussion can be found in: Lovejoy, S., and D. Schertzer (2013), The Weather and Climate: Emergent Laws and Multifractal Cascades, 496 pp., Cambridge University Press, Cambridge.

Figure 3b The axes here are unreadable. Rather than show the obvious (Gaussian assumption makes no sense for dust fluxes), why not consider nonparametric confidence levels, or show at least the log of the dust flux).

Au: We have redone the figure with higher contrast axes/fonts. In fig. 3c (added) we show the same plots after the log transformation. The spikes are less pronounced but are still very far from being Gaussian. We also performed log-log transformations. These resulted in extremes much further from Gaussian at the small scales, but closer to Gaussian at the largest scales. Finally, we added a new probability distribution for the log transformed data (Fig 6b) that showed that the extremes were still power law with low exponents qD: questioning the general habit of log transformation of dust fluxes. As for nonparametric confidence levels, the spike plots already give confidence intervals for the Gaussian, allowing it to be rejected with high levels of confidence. Adding the corresponding plot for the log transformed data would similarly reject the log-Gaussian hypothesis. Our distributions indicate that the power law tail hypothesis would not be rejected, but quantifying this is nontrivial because the theory doesn't indicate the probability level at which the tail is expected to be a power law.

Figure 4 The decrease in power towards the lowest frequencies (>400,000, beta=-2) may well be an artifact: By construction, periods longer than the time-series length divided by two cannot be interpreted, and rules of thumb/good practice is to stick to 1/3rd of the record length. For this 800,000 year-record this would mean that the spectrum could be considered estimable up to timescales of _1/266,000 years. It would further be good practice to subtract at least a linear trend (Chatfield 2016).

Au: Actually even if we only keep the ïĄůïĂǎ= 3 and higher frequencies as the referee suggests, our smooth line is still a pretty good fit to the spectrum. More information can be gleaned about the lower frequencies from fig. 7 that shows that the RMS fluctuations fall off quickly, bounded by ïĄÿ(2)/2 = -1 (i.e. the true ïĄÿ(2)<-2). Since 1+ïĂǎïĄÿ(2), this shows that ïĄć <-1. This analysis is more robust at low frequencies than the spectrum and at least demonstrates that the low frequencies do decay at the lower frequencies. Figure 10 How can the Holocene, being 11,700 years long, have a transition time scale_ïĄťc of 7,900 years?

Au: We used a data segment of 12,500 years and then calculated two fluctuations

for each of the logarithmically spaced scales longer than 6250 years. One of these fluctuations starts at the beginning of the record, the other ends at 12500 years. These two estimates do overlap and thus are noisy. However there are several different scales that are considered and this adds some extra information.

Please also note the supplement to this comment:
https://www.clim-past-discuss.net/cp-2018-110/cp-2018-110-AC2-supplement.pdf